# Prevalence of Varicose Veins and Its Risk Factors among Nurses Working at King Khalid University Hospital Riyadh, Saudi Arabia: A Cross-Sectional Study

**DOI:** 10.3390/healthcare11243183

**Published:** 2023-12-16

**Authors:** Leena R. Baghdadi, Ghadah F. Alshalan, Norah I. Alyahya, Hend H. Ramadan, Abrar M. Alshahrani, Jumana A. Alqahtani, Maha O. Aljarbaa

**Affiliations:** 1Department of Family and Community Medicine, College of Medicine, King Saud University, Riyadh 11362, Saudi Arabia; 2College of Medicine, King Saud University, Riyadh 11362, Saudi Arabia; 441200618@student.ksu.edu.sa (G.F.A.); 441200319@student.ksu.edu.sa (A.M.A.); 441200371@student.ksu.edu.sa (J.A.A.); 441200276@student.ksu.edu.sa (M.O.A.)

**Keywords:** varicose veins, nurses, occupational disease, risk factors, protective factors, occupational health, lifestyle factors

## Abstract

This quantitative observational cross-sectional study assessed the prevalence and level of risk scores for varicose veins among nurses, and the association between varicose veins and sociodemographic, occupational, and lifestyle risk factors. Using simple random sampling, from August–December 2022, 250 nurses from different departments at King Khalid University Hospital completed a validated self-administered questionnaire and underwent an observational physical examination. Most nurses (191) had low-risk scores for varicose veins, 46 nurses had moderate-risk scores, and 13 nurses had high-risk scores. From the outpatient clinics, 61.5% of nurses had significant high-risk scores for varicose veins. Those with a statistically significant association had a family history of varicose veins (*p* < 0.001) and other chronic medical conditions (*p* = 0.04). Physical activity, especially race-walking/running (*p* = 0.006), showed a statistically significant association with the varicose veins score. The years as a staff nurse were statistically significant among the occupational risk factors (*p* = 0.003). The adjusted multivariable regression model showed three significant predictors: a positive family history, running/walking, and total years as a staff nurse (*p* < 0.001, *p* = 0.02, and *p* < 0.001, respectively). Nurses working at outpatient clinics, positive family history, years as a staff nurse, and other chronic conditions are risk factors for varicose veins, while race-walking/running is a protective factor.

## 1. Introduction

The worldwide prevalence of lower extremity varicose veins is 10% to 30% [1,2]. Swollen, bulging, or enlarged blood vessels that generally develop on the legs and feet are known as varicose veins. They can occur in other parts of the body and are blue or dark purple in color with a lumpy, swollen, or otherwise unattractive appearance. This is one of the world’s most frequent peripheral vascular disorders [3]. Varicose veins are not painful in the early stages. Aesthetics are the primary reason patients seek medical help [4].

The initial sign is discomfort, including tiredness and heaviness in the legs and nocturnal cramps. Swelling of the lateral and medial ankles and dorsum of the foot, dull or burning pain, and paresthesia are some of the symptoms that follow. Spider veins or wiggly shaped tiny swellings under the skin surface are apparent changes on the lower limbs. The swellings are soft and painless, and their size is determined by the position of the limb [4]. Risk factors are high vascular blood pressure from prolonged standing, a sedentary lifestyle, pregnancy, gender, and a positive family history [5]. Obesity is not considered a risk factor by itself, although those with obesity class III reported significant limb symptoms, which are indicative of chronic venous insufficiency [6]. In Saudi Arabia, there are few studies about the prevalence of and risk factors for varicose veins. A cross-sectional study in Riyadh showed a high prevalence of varicose veins (47.6%) among 380 females, with associated factors such as increased age, positive family history, high body mass index (BMI), educational level, the number of pregnancies, and use of oral contraceptives [7]. Other studies focused on occupational risk factors for varicose veins in jobs that require standing for long periods of time. In Abha, Saudi Arabia, 42% of teachers had varicose veins, and women had a greater prevalence (37%) compared to men (14%). According to the study, primary school teachers were 39.9% more likely to develop varicose veins than middle- and high-school teachers [8]. A Taiwanese study identified hairdressers as another risk group for varicose veins and focused on long periods of standing at work. They reported that 24.2% of participants had lower extremity varicose veins. For hairdressers ≥45 years old, occupational effects were the major risk factor for varicose veins, while family history was a risk factor for hairdressers ≤45 years old [9]. A study about healthcare workers in Italy found a higher prevalence of venous pathology in nurses and auxiliaries (54.5%) compared to administrative staff (31.6%) and a greater prevalence among females (46.8%) compared to males (24.6%). Therefore, the end results suggest that the combined effects of occupational risk factors like prolonged standing plus other risk factors such as age, family history, and number of pregnancies are likely to cause varicose veins [10]. Some studies focused on nurses as a group at higher risk of varicose veins, as they stand for long periods during their duty hours. A recent study in Jazan revealed a low prevalence of varicose veins among nurses working at King Fahad Central Hospital and Prince Muhammad bin Nasser Hospital. Risk factors found to have statistically significant associations with varicose veins were ethnicity, lifting heavy objects, lack of exercise, family history, use of hormonal therapy, use of contraceptive pills, type of delivery, and parity [11]. A study in Riyadh showed that varicose veins were prevalent in 11% of the total participants (366). There were 40 (39 females and one male) participants with varicose veins, which suggests a low prevalence of varicose veins among nurses working in four separate departments (dermatology, intensive care unit, general surgery, and emergency), with no significant relation between the nature of each department’s practice and the prevalence of varicose veins. Positive family history, age, marital status, long shifts, and lifting heavy items were all significant risk factors for the development of varicose veins among nurses [12]. On the other hand, a study in Egypt revealed that 18.4% (37 out of 201) of nurses have lower-limb varicose veins. A higher prevalence of varicose veins was found among employees in emergency and intensive care units (ICU)/operation rooms who were employed for ≥5 years with >6 daily working hours. The incidence of varicose veins is higher among nurses with chronic constipation, on oral contraceptives, or having had ≥3 pregnancies [13]. In an Iranian study that had nurses with ≥2 years of service from three general hospitals, it was concluded that varicose veins were prevalent in varying degrees in 72.4% of the nurses, with a higher incidence in women than men [14]. Standing for long periods of time was found to be a strong risk factor for varicose veins in 46% of nurses at Nepal’s Dhulikhel hospital, with the highest prevalence in the teaching faculty and the lowest in the orthopedic ward [15]. In India, a study evaluated the risk of varicose veins in nurses working in the ICU and other departments. Nurses working in critical care had a higher risk (9.78) than nurses working in general wards (5.18). The study found that there was an association with other risk factors like age, duration of duty, years of working, and sources of information about varicose veins [16]. In the Udaipur study, 24.17% of 364 nurses had varicose veins, and female nurses had a higher prevalence than males [17]. A Korean study found that the prevalence of varicose veins was 16.2% among nurses at different departments. Nurses working in the operating room had the highest occurrence (36.4%) of varicose veins despite wearing compression stockings, followed by those working in an outpatient clinic (26.9%). No sex-related differences were found in this investigation [18]. In East China, a study among female nurses reported that the prevalence of varicose veins was 32.4% of the total population. Furthermore, varicose veins in the lower extremities show a linear association with age and working years [19,20].

Although there are a few studies on the prevalence of varicose veins among healthcare workers in Saudi Arabia, there is a lack of evidence about the severity of developing varicose veins and its link to several predisposing factors, such as lifestyle and occupational health risk factors. Understanding the association between factors predisposing nurses to severe levels of varicose veins will help in raising public awareness about occupational disorders in our culture, lowering the incidence, or at the very least, delaying the onset or preventing the problem from deteriorating. The recognition of the significance of lifestyle modification, in light of the Saudi Vision 2030, is indicative of a fundamental understanding of the imperative need to promote health and well-being amongst the populace by focusing on promoting preventive care. With a steadfast commitment to enhancing the standard of living and increasing the average life expectancy from 74 years to 80 years, the Vision envisions transformative measures aimed at achieving these objectives [21]. In line with the 2030 Saudi Vision, it is crucial to address the health concerns of nurses, including the prevalence of varicose veins [22]. Therefore, the aim of this study was to assess the level of risk scores for varicose veins among nurses and to find out the association between the levels of scores for varicose veins and several risk factors, including sociodemographic, lifestyle, and occupational risk factors.

## 2. Materials and Methods

A quantitative observational cross-sectional study was conducted among nurses working at King Khalid University Hospital (KKUH) in Riyadh, Saudi Arabia. The study was conducted from August 2022 to December 2022. Data were collected from nurses working in different departments using a validated self-administered questionnaire and observational physical examination [16]; content validity was assessed in their study by distributing it to experts in medicine, surgery, and medical-surgical nursing. Adjustments were made based on the experts’ opinions and suggestions. To determine its reliability, the varicose vein risk assessment tool was computed using the Karl Pearson correlation coefficient (Inter-Rater method). The computed reliability of the tool was r = 0.99, indicating that the tool is reliable.

### 2.1. Study Participants

The sample size was calculated based on a previously published study using the prevalence of varicose veins [13]. By using n = (zα/2)^2^ × *p*(1 − *p*)/d^2^ equation where *p* = 0.184, d = 5%, zα for 95% = 1.96, n= (1.96)^2^ × 0.184 (1–0.184)/0.05^2^ =230.7~231, and assuming a non-response is 10%, the sample size is equal to ~255. A *p* value < 0.05 is considered statistically significant.

A total of 255 nurses of various ages and nationalities were recruited from different departments at KKUH. Nurses who were pregnant, had undergone treatment for varicose veins, and worked for <1 year (n = 5) were excluded from the study. Thus, the final number of recruited nurses was n= 250.

### 2.2. Data Collection Process

The simple random sampling technique was used to collect data daily for one month in August 2022 during the working hours of the inpatients, outpatients, emergency, and ICU departments. A list of potential participants was obtained from the nurse databases provided by the King Saud University Medical City (KSUMC) Nursing Affairs Research and Innovation Department. Each individual on the list was assigned a unique identification number. Using a random number generator, 250 participants were selected from the population list. The eligible participants were contacted when the study’s purpose was explained and were invited to voluntarily participate in the study.

Our data collection included two parts: an English self-administered demographic questionnaire modified by the researchers, which was divided into four parts (sociodemographic information, medical conditions, lifestyle, and occupational risk factors), and a validated varicose vein assessment tool based on an observational physical examination [16]. It consists of 23 questions, and point scores (correct answer was given a score of 1 and the wrong answer was given a score of 0) were assigned to each question. The total maximum score was 23, and the levels of the scores were categorized as low (score 1–7), moderate (score 8–16), and high (score 17–23). The physical examination was conducted by the same researcher after receiving training from a consultant vascular surgeon at KKUH. The study was approved by the King Saud University-College of Medicine Institutional Review Board (Registration No. of the study: E-22-6963). Nurses who agreed to participate gave informed written consent before they were administered the questionnaire. The data collection procedure included an interview via a questionnaire and an observational checklist to assess the likelihood of the participants developing various veins. The checklist has two sections. The first section of the observational checklist evaluated the presence of swelling, tenderness of the lower legs, hardening, change in color, bulging, enlargement or dilatation of the vein, the presence of a spider web, redness, thrombophlebitis, thickness or dryness of the lower legs, and ulceration or irregular pigmentation at or around the ankle. The second section evaluated the presence of fatigue, pitting edema, itching around one or more veins, heaviness or throbbing of the legs, burning sensations, pain or muscle cramping of the lower leg, inflammation of the skin, and restless legs syndrome.

### 2.3. Data Analysis

Data were analyzed using SPSS v.24.0 (IBM, Chicago, IL, USA, US statistical software). Descriptive statistics (mean, standard deviation, frequencies, and percentages) were used to describe the quantitative and categorical variables. A suitable Chi-square test and Fisher’s exact test were used to conduct bivariate statistical analyses. Multivariable-adjusted binary logistic regression was used to predict a moderate/high risk of varicose veins. The statistical significance and precision of the data were reported using a *p*-value of <0.05 and a 95% confidence interval (CI).

## 3. Results

A total of 250 nurses participated in the study, including 235 female nurses (94.0%) and 15 male nurses (6.0%). Their mean age was 36.54 years (±7.73 years), and their mean BMI was 26.6 kg/m^2^ (±4.27 kg/m^2^). The majority of participating nurses had worked for 6–10 years (31.2%). More than half (68.8%) of the nurses were married, and all nurses resided in Riyadh. They mostly lived in the north (43.6%) and center of Riyadh (43.2%). The nurses’ educational levels ranged from a diploma to a master’s degree and doctorate, and 72.0% of the nurses had a bachelor’s degree. The nurses’ information about varicose veins came from a variety of sources, and many of them listed multiple resources. The highest percentages of information came from books (76.8%), healthcare personnel (74.8%), and the internet (74.8%). Table 1 shows the distribution of sociodemographic characteristics of nurses in relation to the departments where they work.

Most nurses (n = 191) had low-risk scores for varicose veins, 46 nurses had moderate-risk scores, and 13 nurses had high varicose vein risk scores (74.4%, 18.4%, and 5.2%, respectively). The number of nurses working in the inpatient departments was 74 (29.6%), those in outpatient clinics were 90 (36.0%), those in the emergency department were 45 (18.0%), and those in the ICU were 41 (16.4%). Among the four departments, the highest prevalence of high-risk scores for varicose veins was found among nurses working in outpatient clinics, followed by inpatient departments (61.5% vs. 23.1%, *p* = 0.03) (Graph 1). Among the sociodemographic and medical factors, there were statistically significant associations with a positive family history of varicose veins and other chronic medical conditions (*p* < 0.001, and *p* = 0.04, respectively) (Table 2).

According to lifestyle factors, physical activity, especially race-walking/running showed a statistically significant preventive association with varicose vein scores (*p* < 0.006) (Table 3). Regarding occupational risk factors, the total years of experience as a staff nurse showed a statistically significant result (*p* < 0.003) (Table 4). Three significant predictors remained in the multivariable-adjusted regression model. The most influential predictor of moderate/high risk of varicose veins is a family history of varicose veins. Participants who reported family history are 3.57 times more likely to have a moderate/high risk of varicose veins, controlling for other factors. Race-walking/running as physical activity is associated with a significant reduction in varicose vein risk; those who race-walk or run for exercise have 4.35 times lower odds of moderate/high risk compared to those who do not do these activities, controlling for other factors. The total experience of 6–10 years in nursing is associated with significantly lower odds of varicose veins compared to 1–5 years of experience, which have 6.25 times lower odds, controlling for other factors (Table 5).

## 4. Discussion

To the best of our knowledge, this is one of the few studies in Saudi Arabia to assess the level of risk scores of varicose veins among nurses in relation to occupational health factors. In our study, we estimated the prevalence of varicose veins, focusing on the level of its risk scores and its associated risk factors among nurses working at different departments of KKUH, by using a simple random sampling technique. We studied sociodemographic risk factors (age, gender, family history, smoking status, etc.), occupational risk factors (longer work history, i.e., employment history), longer working hours, nature of work, and lifestyle factors (weight, diet, and physical activity).

We found that the highest prevalence of high-risk scores for varicose veins was in nurses working in outpatient clinics, followed by those in inpatient departments (61.5% vs. 23.1%, *p* = 0.03). These results were supported by a Korean study, where the prevalence of varicose veins was significantly higher among outpatient clinic and operating room nurses [18]. This could be attributed to the difference in the number of participants recruited from each department. Our study had 74 (29.6%) participants from inpatient departments and 90 (36.0%) participants from outpatient clinics. An Iranian study reported the highest prevalence of varicose veins compared to other countries, where 72.4% of nurses with ≥2 years of service from three general hospitals had lower-limb varicose veins [14]. This could be attributed to the difference in workloads between Iranian hospitals and hospitals in other countries; additionally, it was a questionnaire-based study.

A retrospective Taiwanese cohort study reported that the cumulative incidence of varicose veins among physicians, non-physician health care providers (HCPs), and the general population was 0.12%, 0.13%, and 0.13%, respectively, over a 5-year period. Despite working long hours, physicians in Taiwan had no greater risk of varicose veins than non-physician HCPs or the general population of the study. There was also no significant difference in the varicose vein risk among the various specialist physicians. In comparison to female non-physician HCPs, female physicians appear to have a lower risk of varicose veins. Male physicians had a higher incidence of varicose veins than male non-physician HCPs [20]. However, most of these studies have some limitations, such as selection bias [7] and measurement bias [8,9].

Most of the local and international studies have some limitations, such as a selection bias in the study conducted on females in Riyadh because of the use of a convenience sampling technique [7]. All participants did not undergo a physical examination for their varicose veins in Abha, Saudi Arabia, or in the Taiwanese hairdressers’ studies, which could affect the accuracy of the results [8,9]. The Italian study had a small sample size and potential recall biases [10]. The diagnosis of varicose veins was self-reported, which could be a source of recall bias in the Riyadh and Dhulikhel hospitals [12,15]. Limited sample sizes could not verify cause and effect linkages in the Egyptian and Italian studies [10,13]. Researchers in an Iranian study were unable to investigate the impact of nurses’ lifestyles on nutrition and exercise habits, perhaps due to small sample sizes [14]. The Korean study had some limitations, such as the sample population’s low average age, the short period of service not considered in the study plan, and the number of male participants being too low [18]. The other Taiwanese study weaknesses include the lack of precise information on standing and walking hours, family history, BMI, lower-limb surgery, and parity; all these factors could have influenced the outcomes, and their 5-year follow-up period was too short for the development of outcomes [20].

Sociodemographic, medical, lifestyle, and occupational factors might play a part in developing varicose veins among nurses. Our study found that working at outpatient clinics, a positive family history for varicose veins, total years of work as a staff nurse, and other chronic conditions are risk factors for varicose veins. Race-walking/running is a protective factor. This finding agrees with the study conducted at Riyadh [12] that reported varicose veins were associated with a positive family history (*p*-value = 0.001).

The sociodemographic risk factors for varicose veins that were not statistically significant (*p* > 0.05) were age, gender, BMI, ethnicity, level of education, average family income, smoking, use of hormonal therapy, gravidity, and parity. Age, gender, BMI, use of hormonal therapy, and the number of pregnancies were reported as risk factors for varicose veins in most of the previous studies [12,14]. However, in our study, the majority of nurses (n = 191, 74.4%) had low-risk scores for varicose veins, and 235 (94.0%) participants were females. Although obesity is not considered a risk factor by itself, patients with obesity class III showed significant limb symptoms, including recalcitrant ulcers, prolonged healing, and frequent recurrence of leg ulceration, indicating chronic venous insufficiency [6]. However, in our study, the mean BMI for all nurses was in the lower limit of the overweight range (26.6 kg/m^2^ ± 4.27 kg/m^2^). Chronic diseases studied and other chronic conditions such as polycystic ovarian syndrome, thyroid diseases, and dyslipidemia were statistically significantly associated with varicose veins (*p*-value = 0.004). It is possible that the overweight associated with these conditions increases vein pressure and venous insufficiency, which subsequently leads to varicose vein development [6,7]. On the other hand, among lifestyle factors, race-walking/running was a statistically significant protective factor against developing varicose veins (*p*-value = 0.006), which is in agreement with previously published studies [14,18,19]. Although this protective effect might be explained by the increase in blood supply during the activity, women’s empowerment might explain this association. The 2023 Saudi Vision seeks to empower women in Saudi Arabia by providing them equal opportunities and enabling them to play an active role in the country’s development. Through economic, social, and political empowerment, the government aims to create a more inclusive and prosperous society for all its citizens. Under Saudi Vision 2030, the government has launched several initiatives to encourage physical activity, empower women’s sports, and promote a healthy lifestyle. Some key measures include sports infrastructure, sports events and competitions, sports education, and training, making it easier for individuals to access and engage in physical activity [21].

Our study suggests that occupational factors have a significant impact on the development of varicose veins among nurses. Total years of experience as a staff nurse was a risk factor for varicose veins in our study (*p*-value = 0.003). Similarly, in Udaipur [17], longer work histories were occupational risk factors responsible for lower-limb varicose veins among nurses. It may be explained by the fact that exposure to occupational health education decreases with increased job experience, which might be attributed to the fact that the longer the work experience, the less exposure there is to occupational health education. Although standing during duty hours was not a significant risk factor for varicose veins in our study, this contradicted the finding in the study in Dhulikhel hospital (*p*-value < 0.001) [15]. Sitting during duty hours and night shifts was not statistically significant in this study or previous studies [11,15,23,24,25,26]. Awareness about varicose vein risk factors was not statistically significant, although in the Chinese study it was a protective factor. Therefore, increasing nurses’ awareness is essential for raising the level of protection against varicose veins [19]. Occupational health can play an important role in reducing the risk of varicose veins by educating the nurses and/or raising awareness and maintaining the highest degree of physical well-being among nurses. Raising awareness of occupational safety has been considered a strategy in Australia [27]. In Britain, September has been designated as Vascular Disease Awareness Month [28]. We aspire to have educational courses for the medical staff at KKUH on occupational health, as well as the activation of September as a Vascular Disease Awareness Month campaign. There are three significant predictors adjusted in the multivariable regression model. Positive family history (OR 3.57, 95% CI = 1.82–7.00) and total nursing experience of 6–10 years were associated with a significantly lower risk of varicose veins compared to 1–5 years of experience after controlling other factors of nursing experience (OR 0.16, 95% CI = 0.06–0.46), while running/race-walking showed a protective association (OR 0.23, 95% CI = 0.07–0.80). The 2030 Saudi Vision includes plans to improve employee safety and well-being. The government is focused on enhancing occupational health programs in workplaces. This includes conducting regular health assessments, providing workplace safety training, and implementing preventive measures for occupational hazards and diseases [21,22].

### Strengths and Limitations

Most of the published epidemiological studies conducted on working nurses were weak, with low internal and/or external validity. Several previous studies have some limitations. The current study has a larger sample size of 250 compared to the previous studies [12,14,15], and we used a different assessment tool to assess the level of varicose vein risk scores (mild-, moderate-, and high-risk scores of varicose veins), which is a published tool [16] (demographic pro-forma and varicose veins assessment tool) with a self-administered questionnaire. Additionally, the association between the likelihood of developing varicose veins among nurses and several predictors, including lifestyle and occupational factors, was assessed. Moreover, implementation of standardized protocols for data collection was ensured, which includes comprehensive training of investigators, and can effectively mitigate inter-observer variability in the process of gathering the data. The physical examination was undertaken by the same researcher after receiving training on varicose vein’ examination by a consultant vascular surgeon to minimize measurement bias (i.e., information bias).

This study has a few limitations. Exploring variations in varicose veins across time was constrained by the use of a cross-sectional design. It provides a snapshot of the nurses’ characteristics and their likelihood of developing various veins at a specific point in time by assessing the participants once. While one set of observations per participant in a cross-sectional study might be one of the limitations, it does not necessarily undermine the significance of the study. Cross-sectional studies can still yield invaluable insights, particularly when conducted in conjunction with other research designs or when exploring associations between variables; we examined many associations and conducted multivariable regression analyses. This study was restricted by the use of survey-based and observational examinations without more diagnostic modalities such as Doppler ultrasonography. Although the reliability of the varicose vein risk score was tested (r = 0.99, indicating that the tool was reliable), only the content validity of the questionnaire was completed [16]. While content validity is crucial for ensuring that a questionnaire measures what it intends to measure, reliability is another essential aspect of questionnaire validation, referring to the consistency and stability of the measurement tool [29]. Therefore, it is imperative to conduct future prospective cohort studies to monitor the progress of nurses over an extended period of time and collect multiple observations using Doppler ultrasonography to accurately evaluate the severity of varicose veins. This approach is vital for further scientific exploration and identification of the causal pathways.

## 5. Conclusions

We found that nurses who race-walked and ran for exercise had low-risk scores for varicose veins. Therefore, we encourage nurses to do some physical activity, such as walking, on a daily basis. Nurses who know their scores may change their lifestyle and take preventive actions to improve their condition and prevent their varicose veins from worsening. Knowing the link between nurses and varicose veins will raise public awareness about occupational disorders in our culture, lowering the incidence or, at the very least, delaying the onset or preventing further deterioration. It will change clinical practice by reducing standing time during work hours and helping to focus on the groups at risk during preventive and periodic visits. We recommend future studies on the association between lifestyle factors and varicose veins, particularly diet, because there are few studies on an unhealthy diet as a risk factor for varicose veins and a study on the association between chronic diseases and varicose veins. Finally, we recommend focusing on occupational health by conducting research and increasing awareness, as well as a study to establish the prevalence of varicose veins in Saudi Arabia using a diagnostic tool like Doppler ultrasonography.

## Figures and Tables

**Table 1 healthcare-11-03183-t001:** Sociodemographic characteristics of the nurses by the departments in which they work (N = 250).

Characteristics	Inpatient Dept. n (%)	Outpatient Dept. n (%)	Emergency Dept. n (%)	ICU Dept. n (%)	Total Nurses n (%)
Age (years)	Mean 37.64 ± 6.84	Mean 36.32 ± 8.14	Mean 34.27 ± 6.69	Mean 41 ± 7.86	Mean 36.54 ± 7.53
<30	8 (13.6%)	25 (42.4%)	15 (25.4%)	11 (18.6%)	59 (100.0%)
30–40	45 (34.4%)	46 (35.1%)	24 (18.3%)	16 (12.2%)	131 (100.0%)
>40	21 (35.0%)	19 (31.7%)	6 (10.0%)	14 (23.3%)	60 (100.0%)
Gender					
Females	67 (28.5%)	84 (35.7%)	43 (18.3%)	41 (17.4%)	235 (100.0%)
Males	7 (46.7%)	6 (40.0%)	2 (13.3%)	0 (0.0%)	15 (100.0%)
Nationality					
Saudi	7 (19.4%)	22 (61.1%)	5 (13.9%)	2 (5.6%)	36 (100.0%)
Non-Saudi	67 (31.3%)	68 (31.8%)	40 (18.7%)	39 (18.2%)	214 (100.0%)
Ethnicity					
Arabian	9 (23.7%)	23 (60.5%)	4 (10.5%)	2 (5.3%)	38 (100.0%)
Asian	65 (30.7%)	67 (31.6%)	41 (19.3%)	39 (18.4%)	212 (100.0%)
Marital status					
Single	22 (28.2%)	34 (43.6%)	16 (20.5%)	6 (7.7%)	78 (100.0%)
Married	52 (30.2%)	56 (32.6%)	29 (16.9%)	35 (20.3%)	172 (100.0%)
Level of education					
Diploma in practical nursing	19 (30.6%)	19 (30.6%)	10 (16.1%)	14 (22.6%)	62 (100.0%)
Bachelor’s degree	51 (28.3%)	70 (38.9%)	34 (18.9%)	25 (13.9%)	180 (100.0%)
Master’s degree and doctorate	4 (50.0%)	1 (12.5%)	1 (12.5%)	2 (25.0%)	8 (100.0%)
Total experience as staff nurses					
1–5 years	10 (21.7%)	21 (45.7%)	9 (19.6%)	6 (13.0%)	46 (100.0%)
6–10 years	26 (33.3%)	26 (33.3%)	16 (20.5%)	10 (12.8%)	78 (100.0%)
11–15 years	17 (27.0%)	26 (41.3%)	9 (14.3%)	11 (17.5%)	63 (100.0%)
>15 years	21 (33.3%)	17 (27.0%)	11 (17.5%)	14 (22.2%)	63 (100.0%)
Average family income (SAR)					
<5000	16 (38.1%)	20 (47.6%)	3 (7.1%)	3 (7.1%)	42 (100.0%)
6000–10,000	47 (28.8%)	46 (28.2%)	35 (21.5%)	35 (21.5%)	163 (100.0%)
>10,000	11 (24.4%)	24 (53.3%)	7 (15.6%)	3 (6.7%)	45 (100.0%)
BMI (kg/m^2^)	Mean 26.46 ± 4.37	Mean 25.13 ± 3.86	Mean 24.79 ± 4.67	Mean 26 ± 4.3	Mean 26.6 ± 4.27
Underweight	1 (16.7%)	1 (16.7%)	3 (50.0%)	4 (16.7%)	6 (100.0%)
Normal	27 (23.7%)	48 (42.1%)	23 (20.2%)	16 (14.0%)	114 (100.0%)
Overweight	30 (33.7%)	30 (33.7%)	13 (14.6%)	16 (18.0%)	89 (100.0%)
Obesity	16 (39.0%)	11 (26.8%)	6 (14.6%)	8 (19.5%)	41 (100.0%)
Family history of varicose veins					
Yes	23 (29.9%)	29 (37.7%)	12 (15.6%)	13 (16.9%)	77 (100.0%)
No	51 (29.5%)	61 (35.3%)	33 (19.1%)	28 (16.2%)	173 (100.0%)
Information about varicose veins from					
Family members	23 (29.1%)	31 (39.2%)	14 (17.7%)	11 (13.9%)	79 (100.0%)
Friends and peers	26 (22.4%)	48 (41.4%)	19 (16.4%)	23 (19.8%)	116 (100.0%)
Healthcare personnel	49 (26.2%)	71 (38.0%)	35 (18.7%)	32 (17.1%)	187 (100.0%)
Books	56 (29.2%)	73 (38.0%)	30 (15.6%)	33 (17.2%)	192 (100.0%)
Journals	38 (32.2%)	38 (32.2%)	23 (19.5%)	19 (16.1%)	118 (100.0%)
Newspaper	23 (30.7%)	27 (36.0%)	13 (17.3%)	12 (16.0%)	75 (100.0%)
Internet sources	47 (25.1%)	69 (36.9%)	36 (19.3%)	35 (18.7%)	187 (100.0%)
Social media (Twitter, Facebook, etc.)	36 (29.8%)	45 (37.2%)	19 (15.7%)	21 (17.4%)	121 (100.0%)
Smoking					
Yes	4 (21.1%)	11 (57.9%)	3 (15.8%)	1 (5.3%)	19 (100.0%)
No	70 (30.3%)	79 (34.2%)	42 (18.2%)	40 (17.3%)	231 (100.0%)
Varicose vein risk score					
Low risk	64 (33.5%)	64 (33.5%)	35 (18.3%)	28 (14.7%)	191 (100.0%)
Moderate risk	7 (15.2%)	18 (39.1%)	8 (17.4%)	13 (28.3%)	46 (100.0%)
High risk	3 (23.1%)	8 (61.5%)	2 (15.4%)	0 (0.0%)	13 (100.0%)
Total	74 (29.6%)	90 (36.0%)	45 (18.0%)	41 (16.4%)	250 (100%)

BMI: Body mass index; Dept.: Department; ICU: Intensive care unit; SAR: Saudi Arabian riyal.

**Table 2 healthcare-11-03183-t002:** Varicose vein risk scores and their associations with sociodemographic factors and medical conditions.

Variables	Low Risk	Moderate/High Risk	Statistical Test	*p*-Value
Age (years)			3.12	0.21
<30	42 (71.2%)	17 (28.8%)		
30–40	106 (80.9%)	25 (19.1%)		
>40	43 (71.7%)	17 (28.3%)		
Gender			0.93 ^F^	0.53
Females	178 (75.7%)	57 (24.3%)		
Males	13 (86.7%)	2 (13.3%)		
Ethnicity			0.71	0.40
Arabian	27 (71.1%)	11 (28.9%)		
Asian	164 (77.4%)	48 (22.6%)		
Level of education			1.57 ^F^	0.46
Diploma in practical nursing	44 (71.0%)	18 (29.0%)		
Bachelor’s degree	141 (78.3%)	39 (21.7%)		
Master’s degree and doctorate	6 (75.0%)	2 (25.0%)		
Average family income (SAR)			0.73	0.70
<5000	34 (81.0%)	8 (19.0%)		
6000–10,000	124 (76.1%)	39 (23.9%)		
>10,000	33 (73.3%)	12 (26.7%)		
BMI (kg/m2)			1.37 ^F^	0.74
Underweight	4 (66.7%)	2 (33.3%)		
Normal	90 (78.9%)	24 (21.1%)		
Overweight	67 (75.3%)	22 (24.7%)		
Obesity	30 (73.2%)	11 (26.8%)		
Parity			0.03	0.86
Nulliparous	85 (75.2%)	28 (24.8%)		
Multipara	93 (76.2%)	29 (23.8%)		
Gravidity			0.10	0.76
Nulligravida	76 (76.8%)	23 (23.2%)		
Multigravida	102 (75.0%)	34 (25.0%)		
Use of hormonal therapy			2.38 ^F^	0.50
Hormonal therapy	6 (66.7%)	3 (33.3%)		
Oral contraceptives	16 (76.2%)	5 (23.8%)		
Hormonal therapy and oral contraceptives	2 (50.0%)	2 (50.0%)		
None	154 (76.6%)	47 (23.4%)		
Family history of varicose veins			14.56	<0.001 *
Yes	47 (61.0%)	30 (39.0%)		
No	144 (83.2%)	29 (16.8%)		
Smoking			0.70 ^F^	0.58
Yes	16 (84.2%)	3 (15.8%)		
No	175 (75.8%)	56 (24.2%)		
Diabetes mellitus			0.00 ^F^	1.00
Yes	13 (76.5%)	4 (23.5%)		
No	178 (76.4%)	55 (23.6%)		
Hypertension			1.40	0.24
Yes	16 (66.7%)	8 (33.3%)		
No	175 (77.4%)	51 (22.6%)		
Coronary heart diseases			n/a	n/a
Yes	0 (0%)	0 (0%)		
No	191 (76.4%)	59 (23.6%)		
Heart failure			n/a	n/a
Yes	0 (0%)	0 (0%)		
No	191 (76.4%)	59 (23.6%)		
Deep vein thrombosis			0.78 ^F^	0.42
Yes	1 (50.0%)	1 (50.0%)		
No	190 (76.6%)	58 (23.4%)		
Chronic constipation			0.16 ^F^	0.56
Yes	2 (66.7%)	1 (33.3%)		
No	189 (76.5%)	58 (23.5%)		
Asthma			3.56	0.06
Yes	16 (61.5%)	10 (38.5%)		
No	175 (78.1%)	49 (21.9%)		
Hyperthyroidism			0.76 ^F^	0.34
Yes	3 (60.0%)	2 (40.0%)		
No	188 (76.7%)	57 (23.3%)		
Hypothyroidism			3.13 ^F^	0.13
Yes	10 (58.8%)	7 (41.2%)		
No	181 (77.7%)	52 (22.3%)		
Obesity			0.00	0.95
Yes	20 (76.9%)	6 (23.1%)		
No	171 (76.3%)	53 (23.7%)		
Kidney Failure			n/a	n/a
Yes	0 (0%)	0 (0%)		
No	191 (76.4%)	59 (23.6%)		
Cancer			1.58 ^F^	0.59
Yes	5 (100%)	0 (0%)		
No	186 (75.9%)	59 (24.1%)		
# Other chronic conditions			5.29 ^F^	0.04 *
Yes	4 (44.4%)	5 (55.6%)		
No	187 (77.6%)	54 (22.4%)		

BMI: Body mass index; ^F^: Fisher’s exact test was used when expected counts were less than 5; * Indicates statistically significant *p*-values (*p* < 0.05); n/a: not applicable; # Other chronic conditions: polycystic ovarian syndrome, thyroiditis, migraine, epilepsy, dyslipidemia, incompetent saphenous right and left veins, irritable bowel syndrome, and supraventricular tachycardia; SAR: Saudi Arabian riyal.

**Table 3 healthcare-11-03183-t003:** Prevalence of varicose vein risk scores and their associations according to lifestyle factors.

Variables	Low Risk	Moderate/High Risk	Statistical Test	*p*-Value
Dietary variables				
Caffeinated drinks (coffee, tea, energy drinks)			0.60 ^F^	0.91
Never	11 (73.3%)	4 (26.7%)		
1–3 times per week	42 (77.8%)	12 (22.2%)		
4–6 times per week	24 (72.7%)	9 (27.3%)		
Everyday	114 (77.0%)	34 (23.0%)		
Sugar-sweetened drinks (soft beverages)			2.04 ^F^	0.57
Never	40 (81.6%)	9 (18.4%)		
1–3 times per week	106 (76.8%)	32 (23.2%)		
4–6 times per week	16 (76.2%)	5 (23.8%)		
Everyday	29 (69.0%)	13 (31.0%)		
Donuts, cakes, candy, and chocolate			1.47 ^F^	0.72
Never	38 (76.0%)	12 (24.0%)		
1–3 times per week	126 (76.8%)	38 (23.2%)		
4–6 times per week	15 (83.3%)	3 (16.7%)		
Everyday	12 (66.7%)	6 (33.3%)		
Fast food			0.28 ^F^	0.98
Never	35 (76.1%)	11 (23.9%)		
1–3 times per week	134 (76.6%)	41 (23.4%)		
4–6 times per week	14 (77.8%)	4 (22.2%)		
Everyday	8 (72.7%)	3 (27.3%)		
# Physical activities variables				
Leisure physical activity			7.82 ^F^	0.01 *
Yes	184 (78.3%)	51 (21.7%)		
No	7 (46.7%)	8 (53.3%)		
Light-intensity physical activity			0.61	0.44
Yes	19 (70.4%)	8 (29.6%)		
No	172 (77.1%)	51 (22.9%)		
Moderate physical activities			1.27	0.26
Yes	113 (79.0%)	30 (21.0%)		
No	78 (72.9%)	29 (27.1%)		
Vigorous physical activities			0.13	0.72
Yes	99 (77.3%)	29 (22.7%)		
No	92 (75.4%)	30 (24.6%)		
Race-walking/Running			9.25 ^F^	0.006*
Yes	185 (78.4%)	51 (21.6%)		
No	6 (32.9%)	8 (57.1%)		
Running			4.68	0.03 *
Yes	60 (85.7%)	10 (14.3%)		
No	131 (72.8%)	49 (27.2%)		
Swimming			0.01 ^F^	1.00
Yes	7 (77.8%)	2 (22.2%)		
No	184 (76.3%)	57 (23.7%)		

^F^ Fisher’s exact test was used when expected counts were less than 5; * Indicates statistically significant *p*-values (<0.05); # Physical activities were classified based on the World Health Organization 2020 guidelines on physical activity and sedentary behavior: Leisure physical activity: activity performed by an individual that is not required as an essential activity of daily living and is performed at the discretion of the individual. Examples include recreational activities such as going for a walk and gardening; Light-intensity physical activity: on a scale relative to an individual’s personal capacity, light-intensity physical activity is usually 2–4 on a rating scale of perceived exertion scale of 0–10. Examples include slow walking, bathing, or other incidental activities that do not result in a substantial increase in heart rate or breathing rate; Moderate-intensity physical activity: on a scale relative to an individual’s personal capacity, such as walking briskly (3 miles per hour or faster, but not race-walking), water aerobics, and bicycling slower than 10 miles per hour; Vigorous physical activities: on a scale relative to an individual’s personal capacity, it is usually a 5 or above on a scale of 0–10, such as jogging or race-walking or running, swimming, jumping rope, and soccer.

**Table 4 healthcare-11-03183-t004:** Prevalence of varicose vein risk scores and their variation according to occupational risk factors.

Occupational Risk Factors	Low Risk	Moderate/High Risk	Statistical Test	*p*-Value
Total experience as staff nurse			13.76	0.003 *
1–5 years	29 (63.0%)	17 (37.0%)		
6–10 years	70 (89.7%)	8 (10.3%)		
11–15 years	44 (69.8%)	19 (30.2%)		
>15 years	48 (76.2%)	15 (23.8%)		
Total day duty/month			0.35 ^F^	0.94
1 week	7 (87.5%)	1 (12.5%)		
2 weeks	20 (76.9%)	6 (23.1%)		
>2 weeks	164 (75.9%)	52 (24.1%)		
Total night duty/month			0.44	0.80
None	77 (75.5%)	25 (24.5%)		
2 weeks	22 (81.5%)	5 (18.5%)		
>2 weeks	92 (76.0%)	29 (24.0%)		
Lifting heavy objects			0.79	0.37
Yes	104 (74.3%)	36 (25.7%)		
No	87 (79.1%)	23 (20.9%)		
Consulted an occupational therapist			0.00	0.97
Yes	32 (76.2%)	10 (23.8%)		
No	159 (76.4%)	49 (23.6%)		
Wear crepe bandages stockings			0.10	0.75
Yes	26 (74.3%)	9 (25.7%)		
No	165 (76.7%)	50 (23.3%)		
Hours spent sitting during duty hours			1.31	0.52
<2 h	88 (73.9%)	31 (26.1%)		
2–4 h	84 (80.0%)	21 (20.0%)		
≥4 h	19 (73.1%)	7 (26.9%)		
Hours spent standing during duty hours			3.17 ^F^	0.19
<2 h	14 (93.3%)	1 (6.7%)		
2–4 h	34 (81.0%)	8 (19.0%)		
≥4 h	143 (74.1%)	50 (25.9%)		
Hours spent walking during duty hours			1.08	0.58
<2 h	19 (82.6%)	4 (17.4%)		
2–4 h	46 (79.3%)	12 (20.7%)		
≥4 h	126 (74.6%)	43 (25.4%)		

^F^ Fisher’s exact test was used when expected counts were less than 5; * Indicates statistically significant *p*-values (*p* < 0.05).

**Table 5 healthcare-11-03183-t005:** Multivariable-adjusted binary logistic regression predicting moderate/high risk of varicose veins.

Predictive Factors	OR (95% CI)	*p*-Value
Family history of varicose veins		<0.001 *
Yes	3.57 (1.82–7.00)	
No	Reference	
Other chronic conditions		0.23
Yes	2.53 (0.55–11.67)	
No	Reference	
Race-walking/Running		0.02 *
Yes	0.23 (0.07–0.80)	
No	Reference	
Total experience as staff nurse		
1–5 years	Reference	
6–10 years	0.16 (0.06–0.46)	<0.001 *
11–15 years	0.77 (0.32–1.89)	0.57
>15 years	0.65 (0.26–1.62)	0.35

Regression model adjusted for age, gender, family history of varicose veins, chronic conditions, walking, and total experience of nurses; * Indicates statistically significant *p*-values (*p* < 0.05); Other chronic conditions: polycystic ovarian syndrome, thyroiditis, migraine, epilepsy, dyslipidemia, incompetent saphenous right and left veins, irritable bowel syndrome, and supraventricular tachycardia.

## Data Availability

The data presented in this study are available on request from the corresponding author. The data are not publicly available due to patients’ privacy and the Institutional Review Board’s rules and regulations.

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
