# Peer review of "Prevalence of Varicose Veins and Its Risk Factors among Nurses Working at King Khalid University Hospital Riyadh, Saudi Arabia: A Cross-Sectional Study"

_healthcare, 2023, doi:10.3390/healthcare11243183_

Round 1
Reviewer 1 Report
Comments and Suggestions for Authors
The manuscript by Baghdadi and colleagues evaluates the prevalence of varicose veins and its risk factors using a cross-sectional study design. Overall, the manuscript is well-written, and this is an interesting study. I have a few comments that I think could help strengthen the presentation of the methods and results.
-
For the observational checklist part of the data collection procedure, how many observations were made for each participant? Did you have at least two participating nurses or researchers make the evaluation independently? How accurate is the observation if you only have one observation for each participant? If there is more than one observation per participant, what’s the degree of agreement between these observations?
-
Could you describe your variable selection process for your final multivariate binary logistic regression model? Also, please provide the R-squared of the model and the goodness-of-fit test result.
-
In the final multivariate binary logistic regression, there does not appear to be a significant difference between total experience as staff nurse 11-15 years vs. > 15 years. Could you collapse these two age groups? Or could you treat this variable as a continuous variable? Could you provide the overall Wald-chi square test p-value for the independent variables with over 2 categories?
-
Any missing data in your study? Have the 250 nurses responded to all the questions? If not, how did you deal with missing data? How many participants were included in the final regression model?
-
Please mention if you used a two-sided p-value or a one-sided p-value in your 2.3 Data analysis section.
-
Reviewer 2 Report
Comments and Suggestions for Authors
This article managed to assess the prevalence of and risk factors for varicose veins among nurses working at one hospital in Saudi Arabia. I have several concerns:
Introduction
The introduction lacks logical flow. The authors should describe the magnitude of the problem, its consequences, previous work, the research gaps, and the aims with a spotlight on potential merits. Unfortunately, the authors did not fulfill these points. Several previous studies were not cited while a few studies were described with so many details that should have been moved to the discussion section. The authors also did not clarify the need for a new study examining a topic that was heavily investigated in national and international studies.
Methods
1: The authors claimed using a validated questionnaire to collect data about varicose veins (lines 127 and 128) [Ref 16]. Then they claimed that scores were given based on this questionnaire) lines 142 and 143). However, the authors provided no details about this questionnaire or its validation process. The questionnaire and its scoring system were not even described in the study that was cited by the authors [Ref 16]. Further, the original study where the questionnaire came from was published in a non-peer-reviewed journal on a very limited number of Indian nurses and no validation was conducted apart from the very subjective content validation. It is not obvious what is the scientific basis for calculating this score or applying these cut-offs to define the severity of varicose veins. Varicose veins can be categorized clinically and using Doppler based on many scientific criteria. The authors seem that they used none. Further, this alleged questionnaire should have been used in English and the authors of our study should have translated it, but these details were not provided making the entire data collection method illegitimate.
2: No sample size calculation was conducted and without doubt, the number of the included study population is far below the optimal number.
3: The authors claimed to apply a random sampling approach without giving details of this approach.
4: The authors claimed that 250 nurses were selected before excluding pregnant ones. Yet, the overall study population did not change.
5: The authors claimed at one site that a researcher was taught by a vascular surgeon how to diagnose varicose veins (lines 143 and 144) and in another site the observational checklist was evaluated by the participating nurses (lines 150 and 151). Both sentences are contradicted, yet they are invalid anyway.
6: It is also not clear how data about lifestyle and occupational factors were collected. For example, walking was a yes/no question. It is very non-specific. All nurses walk. What is the definition of walking?
Results
1: The number of male nurses is very small and varicose veins are strictly associated with sex. Excluding men from analyses is required.
2: All tables: % should be provided based on vertical categories not horizontal ones, to allow comparisons.
3: Tables 2-4: What is meant by statistical test?
4: There are several categories with inadequate numbers of participants; therefore, the authors should consider merging these categories to obtain statistical power.
5: Multivaiable-adjusted not multivariate-adjusted. They are different.
6: What are the variables included in the regression models? They should be added to the footnotes.
7: Adjusting the results for major risk factors, such as age, is mandatory even if they did not reach statistical significance in univariate analyses.
8: The term (other chronic diseases) is very non-specific.
9: The authors claimed that more than half of nurses reported vigorous physical activities such as jumping ropes, soccer, and bricks while 70% reported moderate physical activities such as water aerobics. What is meant by bricks? Are playing soccer and doing water aerobics very common among women in Saudi Arabia?! Water aerobics is a heavy exercise. They also claimed that more than 50% reported running. These rates are much higher than previous studies reporting very low rates of physical activity in the country and even much higher than rates reported in Japan and Scandinavian countries, where more facilities for physical activity are provided and physical activity is deeply integrated into the culture. Further, previous Saudi studies showed that the prevalence of physical activity is specifically low among women, making these findings irrational.
Discussion
This section was built on invalid results. Still, it failed even to describe these results or offer a new perspective.
Strengths
The author claimed that the major strengths are the large sample size and the validated questionnaire. Both arguments are inadequate. The sample size is inadequate, and the questionnaire was neither validated in the original nor the current studies. The author in this section used [Ref 17] not [16] to refer to the alleged questionnaire. Ref 17 belongs to a study that did not report using such a questionnaire.
Limitations
The study is full of limitations that were not reported by the authors.
Comments on the Quality of English LanguageExtensive editing is needed.
Round 2
Reviewer 2 Report
Comments and Suggestions for Authors
Thank you for sending the manuscript again to me. I still believe that the study has serious methodological errors and should be rejected for the following reasons: 1: The sampling size and approach are incorrect, 2: The method used for VV diagnosis was non-scientific. The claim that one nurse received a short training by a surgeon does not make her/him qualified to make an independent diagnosis for VV apart from detecting its severity, 3: The questionnaire used is not validated and was used once in a study published in a predatory journal without validation and on a very limited sample size, 4: From the responses of the authors, it is clear that they have no experience with questionnaire validation, 5: Given my knowledge of the Saudi society, I can confirm that the prevalence of practicing PA among Saudi women can never exceed 5% in most optimistic scenarios, 6: The claim of the authors that the participating female nurses were extensively playing soccer and water aerobics is a clear red flag that the used data were inconsistent, and 7: The study lacks originality. There are several national and international studies investigating the same issue.
Comments on the Quality of English LanguageNA.
